# An Aux/IAA Family Member, *RhIAA14*, Involved in Ethylene-Inhibited Petal Expansion in Rose (*Rosa hybrida*)

**DOI:** 10.3390/genes13061041

**Published:** 2022-06-10

**Authors:** Yangchao Jia, Changxi Chen, Feifei Gong, Weichan Jin, Hao Zhang, Suping Qu, Nan Ma, Yunhe Jiang, Junping Gao, Xiaoming Sun

**Affiliations:** 1Beijing Key Laboratory of Development and Quality Control of Ornamental Crops, College of Horticulture, China Agricultural University, Beijing 100193, China; yangchao_jia@126.com (Y.J.); chenchangxi@cau.edu.cn (C.C.); gongfeifei1010@163.com (F.G.); jweichan@163.com (W.J.); s040130@cau.edu.cn (N.M.); yunhe.jiang@cau.edu.cn (Y.J.); gaojp@cau.edu.cn (J.G.); 2Flower Research Institute, Yunnan Academy of Agricultural Sciences, Kunming 650205, China; zhanghao7898@sina.com (H.Z.); qsp@yaas.org.cn (S.Q.)

**Keywords:** rose, flower opening, petal expansion, ethylene, Aux/IAA

## Abstract

Flower size, a primary agronomic trait in breeding of ornamental plants, is largely determined by petal expansion. Generally, ethylene acts as an inhibitor of petal expansion, but its effect is restricted by unknown developmental cues. In this study, we found that the critical node of ethylene-inhibited petal expansion is between stages 1 and 2 of rose flower opening. To uncover the underlying regulatory mechanism, we carried out a comparative RNA-seq analysis. Differentially expressed genes (DEGs) involved in auxin-signaling pathways were enriched. Therefore, we identified an auxin/indole-3-acetic acid (Aux/IAA) family gene, *RhIAA14*, whose expression was development-specifically repressed by ethylene. The silencing of *RhIAA14* reduced cell expansion, resulting in diminished petal expansion and flower size. In addition, the expressions of cell-expansion-related genes, including *RhXTH6*, *RhCesA2*, *RhPIP2;1,* and *RhEXPA8*, were significantly downregulated following *RhIAA14* silencing. Our results reveal an Aux/IAA that serves as a key player in orchestrating petal expansion and ultimately contributes to flower size, which provides new insights into ethylene-modulated flower opening and the function of the Aux/IAA transcription regulator.

## 1. Introduction

Rose (*Rosa hybrida*), one of the most important ornamental crops worldwide, has been widely cultivated and hybridized. Ranging from compact bushes to climbing vines, modern rose harbors flowers of diverse sizes, shapes, colors, and fragrances. Larger flowers are valuable in terms of both evolution and the economy. Changes to internal and external cues of the flower opening process can be responsible for alterations in the flower size [1]. However, the regulatory mechanism of flower opening is far from clear.

Flower opening is a highly complex process, involving the substantial growth and development of floral organs, particularly the petals [2,3]. Before flowers open, most petal cell divisions stop [4]. Therefore, petal expansion is mainly driven by the increasing cell size [5]. Cell expansion, a vital factor that shapes the morphology of plant organs and enables their optimal growth in response to environmental cues [6], depends largely on a coordination of cell wall metabolism, cell turgor, and cytoskeletal reorganization [7]. Ethylene can promote flower opening in many ornamental plants [1]. In rose, ethylene accelerates flower opening while inhibiting petal expansion [5,8,9], and its regulation of flower opening is through the upregulation of gene-encoding ethylene receptors [8,10]. *RhETR3,* a rose ethylene receptor gene, is one of the most important genes in mediating ethylene signaling to promote flower opening [11]. In *Arabidopsis*, the ethylene-insensitive mutants *etr1-1*, *ers1*, and *ein2* show larger leaf areas, which is thought to be caused by the increased cell size [12,13,14], while the constitutive ethylene response mutant *ctr1* shows tiny rosette leaves, smaller inflorescence, and elongated gynoecium [15,16]. The molecular mechanisms controlled by ethylene are stringent [17]. Ethylene promotes radial growth of *Arabidopsis* root cells, but inhibits their axial expansion [18]. However, the regulatory mechanism of ethylene-influenced flower opening is not well understood.

Auxin is crucial for plant growth and development and is involved in cellular processes, including cell division and cell expansion [19]. The exogenous application of auxin enhances the rate and angle of flower opening [20]. Variations in the expression of auxin-related genes, such as *ARF8* and *BPEp*, affect cell elongation during petal growth [21,22]. The perception and transduction of auxin signals require the cooperation of several components, and auxin transcriptional responses rely on a derepression mechanism [23,24]. Aux/IAA proteins act as transcription repressors by heterodimerizing with auxin response factors (ARFs), which then activate or inhibit downstream genes [25,26]. In tomato, the downregulation of *SlARF6* and *SlARF8* by micro*RNA167* causes defects of floral development and female sterility [27], while loss-of-function mutant of *SlIAA9* has a prominent effect on leaf patterning and fruit set [28]. In rose, silencing of *RhIAA16* leads to petal abscission [29]. In addition, Aux/IAA participates in the crosstalk of phytohormones. The knock-down of *SlIAA3* shows auxin and ethylene-related phenotypes [30]. In poplar, the RGL1-ARF7-IAA9 module integrates Gibberellin (GA) and auxin signaling to regulate cambial development [31]. However, the detailed regulation mechanism of Aux/IAAs in flower opening remains largely unknown.

In the present study, we found that the response of petals to ethylene is closely associated with the stage of flower opening. On this basis, the phytohormone-related transcription regulators mediating ethylene-inhibited petal expansion were obtained by RNA-seq analysis. As a result, an Aux/IAA transcription regulator gene *RhIAA14* was identified. Meanwhile, the function of *RhIAA14* was characterized by virus-induced gene silencing (VIGS), and several downstream genes were identified. With these investigations, we aimed to gain some insights into the regulatory mechanisms of petal expansion subjected to ethylene. Our results reveal an Aux/IAA that serves as a key regulator in orchestrating petal expansion and ultimately contributes to flower size.

## 2. Materials and Methods

### 2.1. Plant Materials and Treatment

*Rosa hybrida* ‘Samantha’ plants were propagated as previously described with slight modifications [32]. Rose stems with one node were rooted in vermiculite under intermittent mist for about 4 weeks. The cutting seedlings were transferred to plastic pots containing peat and vermiculite (3:1) and grew under the following conditions: Ambient temperature of 22 ± 1 °C, relative humidity of ~60%, and light/dark photoperiods of 16/8 h. Each tested plant was placed randomly to avoid any positional effects. All of the petal samples in this study were taken from the outermost petal whorl of the flower. Each sample was taken from independent plants, and no mixing was performed.

The effects of different concentrations of ethylene on rose flowers were tested previously [8], and constant and repeatable results were obtained with 10 μL/L ethylene. The flower buds at different opening stages were sealed in a 40 L airtight chamber with 10 μL/L ethylene for 24 h. The control group was sealed with air atmosphere. For each stage, three plants were randomly selected for phenotypic observation. The flowers were photographed and the petals were sampled immediately after the ethylene treatment. The petal angle was measured in degrees (°), by testing the angle between the base-tip line of the outermost petals and the center axis of the flower.

### 2.2. RNA Extraction and Quantitative Reverse Transcription PCR

Total RNA was extracted as previously described [33]. The cDNA templates for quantitative reverse transcription (qRT) PCR were conducted with an HiScript^®^ II reverse transcriptase kit (Vazyme, Nanjing, China). The qRT-PCR was performed by the Step One Plus™ Real-Time PCR System (Applied Biosystems, Carlsbad, CA, USA) using the KAPA SYBR^®^ FAST Universal qRT-PCR kit (Kapa Biosystems, Boston, MA, USA). *RhUBI2* was used as an internal control.

### 2.3. RNA-Seq

RNA-seq libraries were sequenced on an Illumina HiSeq4000 system using the paired-end mode. The reads were mapped to the reference genome *R. chinensis* Old Blush Hm r2.0 [34]. The DEGs (fold change ≥ 2 and *p* < 0.05) were selected by R package DEGSeq [35]. Hierarchical clustering, Venn diagramming, and gene ontology (GO) enrichment analysis of DEGs were implemented by the TBtools [36]. The pie chart was generated via GraphPad Prism 9.0 based on GO analysis.

### 2.4. Sequence Analyses 

A sequence alignment of deduced amino acids was performed via ClustalO with the default parameters [37]. Phylogenetic analyses were conducted in MEGA X [38]. The evolutionary history was inferred using the neighbor-joining method [39]. All of the ambiguous positions were removed for each sequence pair (pairwise deletion option). Motif analyses were performed in the MEME suite [40]. The analysis of the conserved domain was performed using the CDD tools of NCBI [41] and the Pfam database [42].

### 2.5. Virus-Induced Gene Silencing

Virus-induced gene silencing (VIGS) was performed as previously described [43,44]. Briefly, a gene-specific fragment of *RhIAA14* (400 bp in length) was constructed into the TRV2 vector. pTRV1, pTRV2, and pTRV2-*RhIAA14* were transformed into *Agrobacterium tumefaciens*. Then, the rose cuttings were inoculated with the *A. tumefaciens* by the vacuum infiltration method and planted in pots for subsequent analysis. The phenotypes were recorded 40–60 d after infiltration.

### 2.6. Microscopic Observation and Cell Counting

The abaxial epidermal (AbE) cells of the petals were observed and counted as previously described [5,45,46]. Briefly, 3 mm-diameter-discs were excised from a non-veined region of the middle part of a petal length. The discs were fixed by formaldehyde-acetic acid (FAA) solution and cleaned with ethanol. AbE cells were observed by optical digital microscopy (IX73, Olympus), and cell counting was performed using Adobe Photoshop software. 

### 2.7. Statistical Analysis

Statistical analysis of the data was calculated by GraphPad Prism 9.0 software. Two groups of data were compared using Mann–Whitney U-test. The mean of multiple groups of data was compared via Duncan’s multiple range test, with *p* < 0.05 considered as significant.

## 3. Results

### 3.1. Ethylene Regulates Flower Opening in a Development-Specific Manner

To investigate the roles of ethylene during flower opening, we observed the phenotypes of flower buds at different opening stages in response to exogenous ethylene. Flowers at three early opening stages were treated with ethylene for 24 h. In unopened floral buds (stage 1), ethylene did not cause apparent changes in the petals when compared to the air control (Figure 1A). When the flowers started to open (stages 2 and 3), ethylene promoted the flower opening (Figure 1A). Ethylene significantly inhibited the flower size at stage 3 and accelerated petal movement at stages 2 and 3 (Figure 1C,D). The changes in the petal size showed that ethylene significantly inhibited petal expansion at stages 2 and 3 (Figure 1B,E). There were no significant differences in the flower diameter at stage 2 after the ethylene treatment, which may have been caused by the compensation of an enlarged petal angle and a reduced petal size. These results suggest that the response of rose petals to ethylene is closely related to the stage of flower opening.

### 3.2. Identification of Ethylene-Responsive Genes in Rose Petals

To identify potential regulators involved in ethylene-regulated petal expansion, we first constructed a transcriptome database of ethylene-treated rose petals at stages 1 and 2. We obtained a total of 11,884 differentially expressed genes (DEGs) in petals, whose expressions were significantly changed during development or after ethylene treatment (fold change ≥ 2 and *p* < 0.05) (Figure 2A and Appendix A). After the ethylene treatment, we found 5505 DEGs present in petals at stage 2 and only 3760 DEGs in petals at stage 1. In addition, from stage 1 to stage 2, the expression levels of 8662 DEGs were significantly changed in the petals. To identify DEGs affected by both development and ethylene (stage 2), a total of 2000 DEGs were roughly screened out (Figure 2B).

To elaborate on the functions of these DEGs, we carried out GO analysis. In the GO category of the biological process, DEGs were significantly enriched in the cell-expansion-related process (water homeostasis, cell wall organization, lignin biosynthesis, and xyloglucan metabolism) and hormone-signaling pathways (response to auxin, abscisic acid, cytokinin, jasmonic acid, salicylic acid, and ethylene) (Figure 2C). Among which, we identified 72 DEGs related to hormone pathways, and the DEGs responsive to auxin were the largest group, including 20 DEGs (Figure 2D). These results suggest that auxin-signaling-related genes may play a crucial role in the flower opening of rose.

### 3.3. RhIAA14 Encodes an Ethylene-Inhibited Transcriptional Regulator

Based on the RNA-seq results, the expression of DEGs in auxin-signaling pathways was robustly regulated by ethylene. Therefore, we focused on transcription factors and regulator genes in the auxin-signaling cascade for further analysis. Among them, the transcript level of *RchiOBHmChr4g0389621* was inhibited by ethylene at stage 2, specifically. *RchiOBHmChr4g0389621* encodes a putative protein containing 237 amino acids. Phylogenetic analysis indicated that RchiOBHmChr4g0389621 belongs to the Aux/IAA family and shares a close relationship (more than 70% similarity) with IAA14 from *Arabidopsis thaliana* (Appendix A). Therefore, we designated it as RhIAA14. To understand the evolutionary conservation of RhIAA14, we performed multi-alignment of IAA14 homologs from different species. The results showed that RhIAA14 is similar to IAA14 proteins from a range of plant species and contains all four conserved domains (DI–DIV) of the Aux/IAA family [45,46] (Appendix A).

To further verify the expression pattern of *RhIAA14*, we carried out qRT-PCR analysis. The results showed that the expression of *RhIAA14* was significantly suppressed by ethylene at stage 2, while it remained relatively constant at stage 1 (Figure 3A). *RhIAA14* expression was increased from stage 1 to 4 and decreased at stage 5 (Figure 3B). In addition, the expression of *RhIAA14* was higher in floral tissues, especially in petals and stamens (Figure 3C), suggesting that *RhIAA14* may play a critical role in petal growth and development.

### 3.4. Silencing of RhIAA14 Inhibits Petal Expansion

To investigate the role of *RhIAA14* in flower opening, we silenced it by VIGS in rose plants. Considering that the expression of *RhIAA14* is relatively constant from stage 3 to 4, we selected petals at stages 3–4 for silencing detection. The qRT-PCR results demonstrate that the expression of *RhIAA14* was significantly reduced in *RhIAA14*-silenced petals (Figure 4A). Compared with the control plants, the silencing of *RhIAA14* resulted in flowers with a smaller size (Figure 4B). The flower diameters and petal angles were measured every other day. From the seventh day, the flower diameters of *RhIAA14*-silenced plants were significantly smaller than those of the TRV controls (Figure 4C). However, the petal angles of the *RhIAA14*-silenced flowers were similar to those of the TRV controls (Figure 4D). These results suggest that *RhIAA14* influences flower size, to a large extent, through the petal size rather than the petal angle.

To confirm whether *RhIAA14* plays a role in petal expansion, the petal size was further analyzed. Compared with the TRV control, the silencing of *RhIAA14* significantly decreased the petal area from the fifth day (Figure 5A,B). The cell expansion of the abaxial epidermal of the petals was directly related to petal expansion [5]. To confirm whether *RhIAA14* influenced the cell expansion of the petals, we took discs at the center of the petals for microscopic observation. The results show that the sizes of the AbE cells were apparently smaller in the *RhIAA14*-silenced petals than in the TRV controls (Figure 5C). In addition, the number of AbE cells per visual field in the *RhIAA14*-silenced petals increased by approximately 25% (Figure 5D). Together, these results indicate that *RhIAA14* influences petal expansion at least partially by regulating the cell size.

### 3.5. Silencing of RhIAA14 Decreases Expression of Cell-Expansion-Related Genes

Cell expansion is subject to a range of cell-expansion-related genes [47,48]. To explore the molecular mechanism by which *RhIAA14* positively regulates cell expansion, we investigated the regulating effects of *RhIAA14* on the expression levels of cell-expansion-related genes. Based on our previous study, we selected several cell-expansion-related genes, including the xyloglucan transferase (*RhXTH6*), cellulose synthase (*RhCesA2*), aquaporin (*RhPIP2;1*), and expansin (*RhEXPA8*) genes [49]. The qRT-PCR results demonstrate that the expression levels of all these genes were significantly altered in the *RhIAA14*-silenced petals. Compared with the TRV control, the silencing of *RhIAA14* decreased the expression of *RhXTH6*, *RhCesA2*, *RhPIP2;1,* and *RhEXPA8* by 70.1%, 49.6%, 77.7%, and 88.5%, respectively (Figure 6). 

## 4. Discussion

The growth and development of plants are restricted by many specific temporal nodes, such as cell proliferation arrest, flowering transition, and ripening/senescence initiation processes [50,51,52,53]. Flower opening is tightly regulated by diverse internal and external cues [54]. In the current study, we found that the response of rose flowers to ethylene varies with developmental stages. Only when flower buds entered stage 2 (flower buds with open sepals) could petals respond morphologically to ethylene by exhibiting a notable change in the petal angle and area. This phenomenon is similar to the onset of ethylene-induced leaf senescence and fruit ripening. Only when leaves or fruits reach sufficient maturity can ethylene lead to premature senescence or ripening [55,56]. Indeed, the RNA-seq analysis showed that rose petals undergo dramatic changes from stage 1 to 2, which may also explain why cut flowers of rose are always harvested after stage 2. However, the molecular nature of the temporal node during flower opening remains largely unknown.

Larger flowers boost pollination and are preferred by most people, as well. For more efficient breeding of large-flowered rose cultivars, the underlying molecular mechanism should be elucidated. Organ size largely depends on the number and size of its constituent cells [50,57,58]. In the current study, we showed that *RhIAA14* works on petal expansion by fine-tuning cell expansion. In fact, after the formation of the flower primordium, all cells of the floral organ enter a continuous division stage, and variations in the timing or rate of cell division can lead to changes in the organ size [59]. Here, due to the limited observation period and local counting of cell numbers, the effect of *RhIAA14* on cell proliferation could not be excluded. It may help us in approaching the truth by detecting the expression levels of genes related to cell proliferation, and conducting a wider observation. 

Several phytohormones largely contribute to similar biological processes and affect flower opening through a series of sequential steps [60,61]. In terms of organ size, ethylene is generally considered a growth inhibitor, with the well-known triple response of etiolated seedlings [15,55]. Our previous study showed that ethylene can reduce the petal size and induce petal movement [5,9]. *RhPIP1;1* and *RhPIP2;1* are involved in ethylene-regulated petal cell expansion [5,56]. The inhibition of cell expansion by ethylene is mediated via a microRNA164-RhNAC100 module that regulates cellulose synthesis and cell turgor [62]. Meanwhile, the RhEIN3-RhGAI1 module mediates ethylene-suppressed cell expansion by controlling the expression of *RhCesA2* [63]. Our recent study found that transcription factor RhNF-YC9 regulates the speed of petal expansion by mediating the crosstalk between GA and ethylene [49]. This evidence demonstrates that the effects of ethylene on cell expansion in rose petals are mediated by sophisticated networks of transcriptional regulation. In this study, we demonstrated that the expression of several transcription factor and regulator genes that are involved in the signaling of auxin, abscisic acid, and cytokinin were dramatically altered during flower opening and under ethylene treatment. Furthermore, an ethylene-inhibited *Aux/IAA* gene, *RhIAA14*, was identified in rose petals. In fact, auxin is a rare supernatural messenger in plants [64]. Auxin-regulated cell expansion is involved in many important developmental processes, such as organ growth [64], tropic bending [65], root hair and apical hook development [66,67], and shoot elongation subject to environmental cues [68,69]. A long-standing controversy has troubled researchers about the relationship between ethylene and auxin [70,71]. In *Arabidopsis*, by stimulating the transcription of *WEI2* and *WEI7*, ethylene triggered the accelerated production of auxin in the roots of seedlings [72]. In rose, during petal abscission, *RhERF1* and *RhERF4* coordinated ethylene and auxin signals to regulate pectin metabolism [73]. In the current study, we propose that the auxin-signaling regulator *RhIAA14* functions as a hub, integrating auxin- and ethylene-signaling in petal expansion, whereas the regulatory mechanism of *RhIAA14* during petal expansion still needs further study. Of course, other growth regulators, such as gibberellic acid, cytokinin, and brassinosteroid, also make a difference in regulating the organ size, and there are dynamic interactions among these hormones. Therefore, the integration of these hormonal signals in regulating petal growth and the molecular mechanisms behind them also require in-depth exploration.

The SCF^TIR1/AFB^-Aux/IAA-ARF signaling modules mediate a majority of auxin-related changes in growth and development of plants [74,75]. Aux/IAA protein degradation is a pivotal event in auxin signaling [76,77]. Specifically, Aux/IAA proteins repress ARF function by separating ARF proteins from their target promoters [24] or recruiting TOPLESS (TPL)/TPL-related (TPR) corepressors, resulting in the inactivation of chromatin and the silencing of ARF target genes [22,23,24,25]. To explore the potential interacting proteins of RhIAA14, we also constructed a protein–protein interaction (PPI) network by means of the STRING database. The results show that RhIAA14 could interact with several ARFs and transport inhibitor response 1/auxin-signaling F-box proteins (TIR1/AFBs) (Appendix A). However, the regulatory mechanism between RhIAA14 and its interacting proteins still needs further investigation.

## 5. Conclusions

In this study, rose petals exhibited a discrete response to ethylene in a development-related manner. With the use of RNA-seq, an Aux/IAA family gene, *RhIAA14*, was identified. The expression of *RhIAA14* is stage-specifically inhibited by ethylene. The silencing of *RhIAA14* reduces the petal size, leading to smaller flowers, which may be mediated by a series of cell-expansion-related genes, including *RhXTH6*, *RhCesA2*, *RhPIP2;1*, and *RhEXPA8*. Our results reveal an auxin-signaling component that serves as a key player in orchestrating cell expansion and ultimately contributes to flower size, which provides gene resources for breeding large-flowered rose cultivars.

## Figures and Tables

**Figure 1 genes-13-01041-f001:**
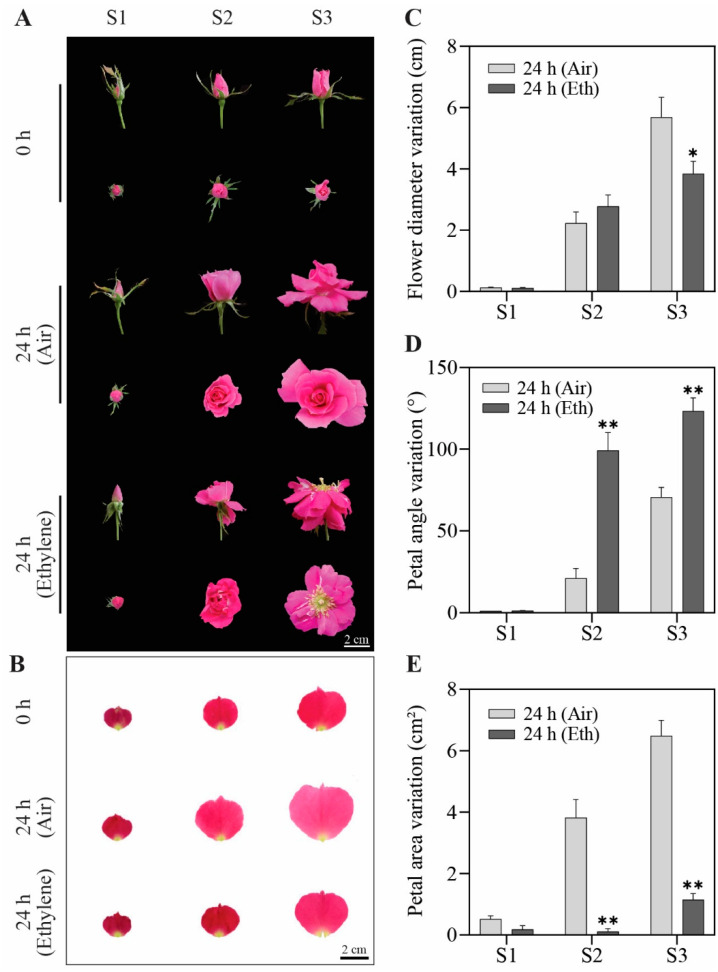
Effect of exogenous ethylene on flower buds at different stages. (**A**) The phenotypes of flower buds at different stages exposed to ethylene or air for 24 h. Stages of flower opening were defined as follows: S1, flower buds with loose sepals (~12-day-old flower buds); S2, flower buds with open sepals (~14-day-old flower buds); S3, flower buds with loose petals (~16-day-old flower buds). First row, side view of flowers; second row, top view of flowers. Bar = 2 cm. (**B**) Petal phenotypes at different stages with air or ethylene treatment for 24 h. Bar = 2 cm. (**C**–**E**) Quantitative statistics of phenotypes. Flower diameter (**C**), petal angle (**D**), and petal area (**E**) were measured. The results are the mean of three biological replicates ± SD, and asterisks indicate significant differences, in accordance with Mann–Whitney U-test (* *p* < 0.05; ** *p* < 0.01).

**Figure 2 genes-13-01041-f002:**
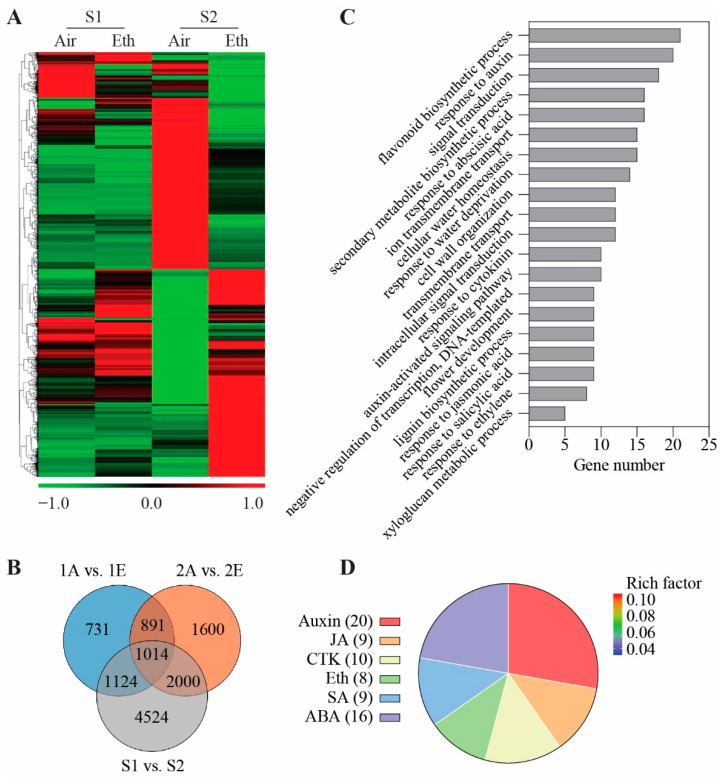
Screening of ethylene-responsive genes in rose petals. (**A**) Hierarchical clustering analysis of DEGs. Petals at stages 1 and 2 exposed to air and ethylene for 24 h. Log_10_ of FPKM in each gene set was used to generate the cluster. Different shades of red, black, and green indicate the extent of the changes, in accordance with the color bar provided. (**B**) Venn diagram analysis of DEGs in different groups. 1, stage 1 (S1); 2, stage 2 (S2); A, air; E, ethylene. (**C**) Gene ontology functional classification analysis of the 2000 DEGs identified from Venn diagram. (**D**) DEGs related to hormone-signaling pathways. The numbers in the parentheses indicate the number of DEGs. Rich factor refers to the ratio of the number of DEGs located in GO term to all of the genes in the same term.

**Figure 3 genes-13-01041-f003:**
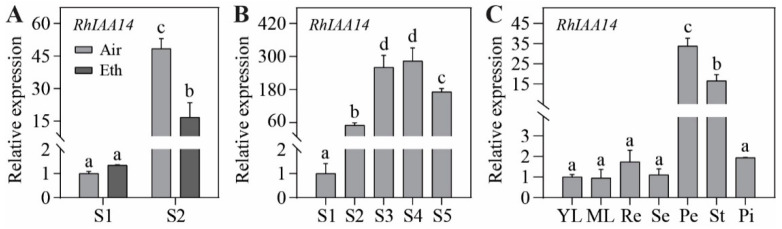
The qRT-PCR analysis of the expression level of *RhIAA14*. (**A**) Expression of *RhIAA14* in rose petals at stages 1 and 2 in response to ethylene. (**B**) Expression of *RhIAA14* in rose petals during different stages of flower opening. Stages of rose flowering were defined as follows: S1, flower buds with loose sepals; S2, flower buds with open sepals; S3, flower buds with loose petals of the outermost layer; S4, flowers with loose petals of inner layers; S5, fully open flowers. (**C**) Expression of *RhIAA14* in leaves and different floral tissues. YL, young leaf; ML, mature leaf; Re, receptacle; Se, sepal; Pe, petal; St, stamen; Pi, pistil. The qRT-PCR was performed using three biological replicates, and the data are presented as the mean ± SD. The letters indicate significant differences, in accordance with Duncan’s multiple range test (*p* < 0.05). *RhUBI2* was used as an internal control.

**Figure 4 genes-13-01041-f004:**
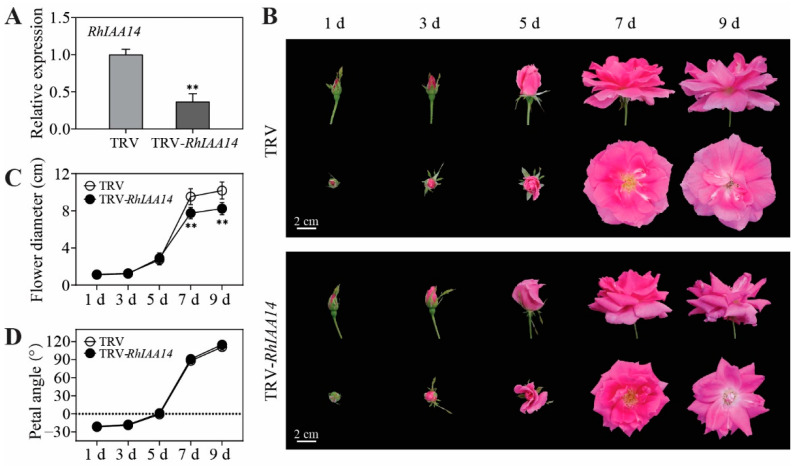
Flower phenotypes of silenced *RhIAA14*. (**A**) Expression of *RhIAA14* in *RhIAA14*-silenced and TRV control petals. TRV, empty vector; TRV-*RhIAA14*, TRV containing a fragment of *RhIAA14*. *RhUBI2* was used as an internal control. (**B**) Phenotypes of TRV and TRV-*RhIAA14* flowers were recorded every other day. Day 1, flower buds with loose sepals. Bar = 2 cm. (**C**,**D**) Quantitative statistics of phenotypes. Flower diameter (**C**) and petal angle (**D**) of *RhIAA14*-silenced plants and TRV controls were measured every other day. The results are the mean of three biological replicates ± SD, and asterisks indicate significant differences, in accordance with Mann–Whitney U-test (** *p* < 0.01).

**Figure 5 genes-13-01041-f005:**
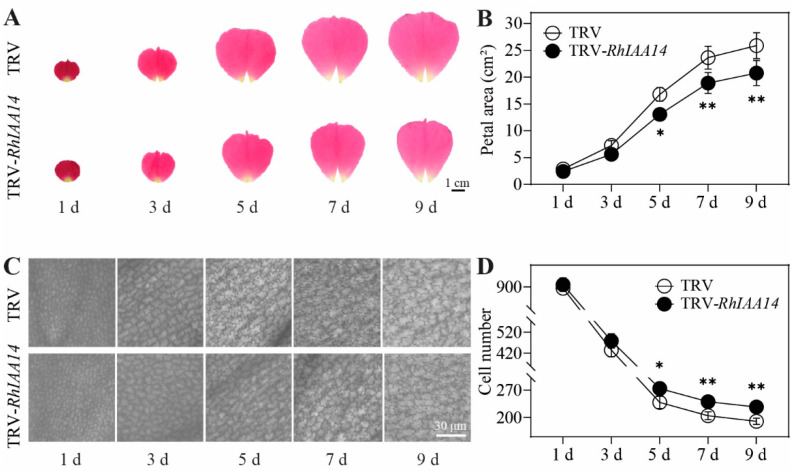
Petal and AbE cell phenotypes of silenced *RhIAA14*. (**A**) Image of flattened petals. TRV, empty vector; TRV-*RhIAA14*, TRV containing a fragment of *RhIAA14*. The images were taken every other day. (**B**) Sizes of *RhIAA14*-silenced petals. Bar = 1 cm. (**C**) Image of AbE cells at the center part of petals. (**D**) AbE cell density of *RhIAA14*-silenced petals. The cells were counted in visual fields of 300 × 300 μm. Bar = 30 μm. The results are the mean of three biological replicates ± SD, and the asterisks indicate significant differences, in accordance with Mann–Whitney U-test (* *p* < 0.05; ** *p* < 0.01).

**Figure 6 genes-13-01041-f006:**
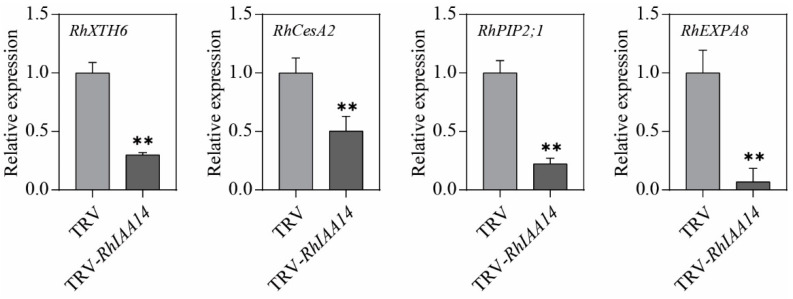
Transcript levels of cell-expansion-related genes in *RhIAA14*-silenced and TRV control petals. Petals at stages 3–4 were used for qRT-PCR analysis. The results are the mean of three biological replicates ± SD, and the asterisks indicate significant differences, in accordance with Mann–Whitney *U*-test (** *p* < 0.01). *RhUBI2* was used as an internal control.

## Data Availability

The data presented in this study are available in the article and Appendix A.

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
