# Peer review of "An Aux/IAA Family Member, RhIAA14, Involved in Ethylene-Inhibited Petal Expansion in Rose (Rosa hybrida)"

_genes, 2022, doi:10.3390/genes13061041_

Round 1
Reviewer 1 Report
Dear Author,
The present manuscript is well executed and properly written. However, there is no details abour RNA-seq analysis in the materials and methods. Though there is stage 1-3 were mentioned, no details about the days of sampling. Like, Stage 1 is how many days old flower bud ? What are the plants used to multiple sequence alignement need to be given in the legend eg., At, Arabidopsis thaliana ... and also need to be expalined in the text, Reason to select those plants ?
Author Response
We greatly appreciate the detailed and thorough reviews, which allowed us to clarify, expand on and improve our manuscript. We have carefully revised the manuscript according to your comments and marked up the changes using the “Track Changes” function in our revised manuscript.
Point 1: There is no details about RNA-seq analysis in the materials and methods.
Response 1: Thanks for your valuable suggestion. We have added the detailed description of RNA-seq analysis in the Materials and Methods section.
Point 2: Though there is stage 1-3 were mentioned, no details about the days of sampling. Like, Stage 1 is how many days old flower bud ?
Response 2: Thanks for your valuable comment. We have added the description of flower buds days at each stage in the figure legend.
Point 3: What are the plants used to multiple sequence alignement need to be given in the legend eg., At, Arabidopsis thaliana ... and also need to be expalined in the text, Reason to select those plants ?
Response 3: Thanks for your valuable suggestion. We have added the full name of each species in the figure legend, and explained the reasons for selecting these species in the results section.

Reviewer 2 Report
This paper is written well and reports novel findings significantly supported by the experimental data. I have some comments and suggestions which can be found directly in a manuscript file (see attached). Some of these comments refer to style or language (it should be noted that this text is generally of fine language quality despite some minor flaws). The only more or less serious concern is about application of Student's t test for comparison of small samples, which is not correct (if I understood the work properly).
I also have some suggestions considering figures, as some of their parts are not easily readable.

Author Response
We greatly appreciate the detailed and thorough reviews, which allowed us to clarify, expand on and improve our manuscript. We have carefully revised the manuscript according to your comments and marked up the changes using the “Track Changes” function in our revised manuscript. Please see the attachment.
Point 1: Some of these comments refer to style or language (it should be noted that this text is generally of fine language quality despite some minor flaws).
Response 1: Thank you for your careful review, we have checked and revised point by point in the revised manuscript.
Point 2: The only more or less serious concern is about application of Student's t test for comparison of small samples, which is not correct (if I understood the work properly).
Response 2: Thanks for your valuable comment. We have modified the error in the revised manuscript.
Point 3: I also have some suggestions considering figures, as some of their parts are not easily readable.
Response 3: Thanks for your suggestion. We have modified some figures to make it easily understand in the revised manuscript.
